# Phenolic Monoterpenes Conversion of *Conobea scoparioides* Essential Oil by Hydrotalcite Synthesized from Blast-Furnace Slag

**DOI:** 10.3390/plants13091199

**Published:** 2024-04-25

**Authors:** Monaliza M. Rebelo, Eloisa Helena A. Andrade, José Augusto M. Corrêa, José Guilherme S. Maia

**Affiliations:** 1Programa de Pós-Graduação em Química, Universidade Federal do Pará, Belém 66075-110, PA, Brazil; monalizamaia@yahoo.com.br (M.M.R.); eloisandrade@ufpa.br (E.H.A.A.); 2Programa de Pós-Graduação em Geologia e Geoquímica, Universidade Federal do Pará, Belém 66075-110, PA, Brazil; jamc@ufpa.br; 3Programa de Pós-Graduação em Ciências Farmacêuticas, Universidade Federal do Pará, Belém 66075-110, PA, Brazil

**Keywords:** pataqueira, thymol, essential oil composition, hydrotalcite, blast-furnace slag

## Abstract

*Conobea scoparioides* (Plantaginaceae) is an herbaceous plant known as “pataqueira” that grows wild in seasonally wet areas of the Amazon region. It is used for aromatic baths and anti-protozoan remedies by the Brazilian Amazon native people. The main volatile compounds identified in the essential oil of “Pataqueira” were the phenolic monoterpenes thymol and thymol methyl ether and their precursors, the monoterpene hydrocarbons α-phellandrene and *p*-cymene. A hydrotalcite synthesized from blast-furnace slag exhibited a 3:2 (Mg/Al) molar ratio, and this layered double hydroxide (LDH) was evaluated as a catalyst in converting the main monoterpenes of the “Pataqueira” oil. This action significantly increased the thymol content, from 41% to 95%, associated with the percentual reduction in other main components, such as thymol methyl ether, α-phellandrene, and *p*-cymene. The LDH reaction showed a strong tendency towards producing hydroxylated derivatives, and its behavior was similar to the hypothetical plant biosynthetic pathway, which leads to the production of the monoterpenes of “Pataqueira” oil. Thymol and its derivatives are potent antiseptics applied in pharmaceutical and hygienic products as antibacterial, antifungal, and antioxidant properties, among others. The present work reports a natural source with a high thymol content in aromatic plants from the Amazon, with evident economic value.

## 1. Introduction

The world scientific community’s attention is increasingly focused on environmental protection and the efficient use of natural resources. Essential oils extracted from aromatic plants containing terpenoids based on multiple isoprene units are inexpensive and renewable raw materials. These oils have biological activities and are widely used in developing new medicines and synthesizing fine chemicals and other intermediates [1]. 

The inventory of aromatic plants from the Amazon and their essential oils and aromas is being performed systematically by a group of researchers working at local scientific institutions [2]. *Conobea scoparioides* (Cham. & Schltdl) Benth. (syn. *Sphaerotheca scoparioides* Cham. & Schltdl.), belonging to Plantaginaceae, is an herbaceous aromatic plant known as Pataqueira or Vassourinha-do-brejo, with occurrence in the bed of small streams and wetlands of the Amazon region. It is used for aromatic baths and anti-protozoan remedies [3,4]. Previously, the main volatile compounds identified in the essential oil of Pataqueira were thymol, methyl thymol, α-phellandrene, and *p*-cymene [5,6]. The plant’s essential oil and methanolic extract have shown significant antioxidant activities [7]. The plant seeds and tiny seedlings adapted well to the hydroponic cultivation system, producing 4 kg/m^2^ biomass during four months of cropping [8].

The layered double hydroxides (LDHs) are anionic clays of natural or synthetic origin, known as compounds of the hydrotalcite type. LDHs are promising materials for heterogeneous catalysis used effectively in various organic reactions (dehydrogenation, dehydration, aldol condensation, polymerization, isomerization, hydrogenation, transesterification, and others) [9,10,11,12]. Hydrotalcite has advantages compared to conventional catalysts: it does not mingle with the reaction products, is easy to handle, and has shown good separation and reuse, contributing to an appropriate sustainable development. Hydrotalcite is widely studied due to its diverse applicability, anion exchange capacity, accessible accommodation of various cations on coverslips, and high basicity [13,14,15]. Hydrotalcite is easily synthesized at a relatively low cost, obtained in high purity and a high degree of structural order, with various properties, and adjusted according to the required synthesis variables [16].

Various forms of use of hydrotalcite as catalysts have been reported, although the studies are concentrated on their mixed oxides. The isomerization of eugenol and safrole over MgAl hydrotalcite was performed by Kishore and Kannan (2002) [17] to obtain isoeugenol and isosafrole, with applications in the fragrance and pharmaceutical industries. Isomerization of α-pinene over dealuminated ferrierite-type zeolites in the liquid phase was conducted by Rachwalik and co-workers (2007) [18], and the selectivity toward the camphene and limonene was close to 85%. Camphene and limonene are intermediate compounds that produce many other monoterpenoids of industrial importance [19].

So, it became attractive to evaluate the hydrotalcite synthesized from blast-furnace slag as a catalyst for monoterpene constituents occurring in some essential oils of the Brazilian Amazon. The main focus of this work was to evaluate the action of hydrotalcite as a catalytic material in the eventual conversion of thymol, thymol methyl ether, α-phellandrene, and *p*-cymene, the main monoterpene components of the essential oil *Conobea scoparioides*.

Thymol has been used for its different biological activities, such as antimicrobial, antioxidant, anti-inflammatory, antibacterial, antifungal, antidiarrheal, anthelmintic, analgesic, digestive, abortifacient, antihypertensive, spermicidal, depigmenting, antileishmanial, anticholinesterase, insecticidal, and many others. This phenolic compound is among the essential scaffolds for medicinal chemists to synthesize more bioactive molecules by further derivatization of the thymol moiety. It is a powerful antiseptic, with applications in pharmaceutical and hygienic products as an antibacterial, antifungal, anti-inflammatory, antioxidant, and others [20,21,22,23,24].

The antioxidant capacity of thymol and thymol methyl ether was examined using oxygen radical absorption capacity and intracellular antioxidant capacity assays. Thymol displays stronger peroxyl radical and hydroxyl radical scavenging capacity and a more reducing capacity than thymol methyl ether, which can explain its hydrogen or electron-donating capacity. However, thymol methyl ether significantly protects against peroxyl radical- and Cu^2+^-induced oxidative stress compared to thymol in the intracellular antioxidant capacity and lipid peroxidation assays using HepG2 cells. These results illustrate thymol methyl ether’s higher cell membrane permeability and transformation to thymol, which results in a significant intracellular antioxidant capacity, contributing to protection against lipid peroxidation [25]. 

The discovery of a natural source with high levels of thymol and thymol methyl ether is a purposeful action to select Amazonian aromatic plants with economic value, as is the present case of Pataqueira. 

## 2. Results and Discussion

### 2.1. Hydrotalcite Characterization

The LDHs synthesized based on the blast-furnace slag, at a temperature of 45 °C, exhibit the carbonate anion and the chloride ion intercalated on the lamellar space and are of the Mg-Cl-Al-CO_3_ type [26]. The carbonate anion was introduced by sequestration of CO_2_ from the atmosphere and deionized H_2_O, while the chloride ion was derived from HCl and MgCl_2_·6H_2_O. These intercalations provide excellent stability to the synthesized LDHs. The X-ray diffractogram (XRPD) of BFS-LDH-4MgAl is shown in Figure 1. It has been found that it is composed of data of interplanar distance (reflections *d*: 7.87, 3.92, 2.61, 1.53, 1.50, 1.43, and 1.32 Å) that are typical of hydrotalcite obtained from blast-furnace slag and comparable to values previously described [16,27,28]. In addition, the value of the basal spacing *d*(003), 7.87 Å, and the half-width (FWHM), 0.43, strengthens the argument that BFS-LDH-4MgAl belongs to the Mg-Al-Cl-CO_3_ system. These values are characteristic of hydrotalcite formed by two anions in the same interlayer space [26].

A semi-quantitative chemical analysis of BFS-LDH-4MgAl showed contents (%) of O: 59.6, Mg: 25.0, Al: 8.8, Cl: 3.6, and C: 1.7 as the main elements, in addition to Si, Mn, and Fe with values less than 1%. This composition characterizes the synthesized hydrotalcite as the Mg-Al-Cl-CO_3_ type. The molar ratio identified in BFS-LDH-4MgAl was Mg/Al 3:2, slightly below the theoretical ratio of 4:0. This difference is likely related to the pH of 11 in the synthesis process, which would be insufficient to lead to precipitation throughout the Mg in the solution [16].

In the SEM micrographs, obtained by emission of secondary electrons, with magnitudes of (a) 2.300× (b) 5.030× (see Figure 2), it is possible to view the uneven surface of BFS-LDH-4MgAl, a typical lamellar porous material, with the presence of varying size aggregates, to less than 1 microns, resulting in ultra-thin particles. The porosity is a feature of a material with good catalytic activity.

### 2.2. Oil Characterization

The individual components of the Pataqueira oils were identified in a GC-MS instrument by comparing mass spectrum and GC retention data with authentic compounds previously analyzed and stored in the data system, as well as with the aid of commercial libraries containing retention indices and mass spectra of volatile compounds commonly found in essential oils [29,30]. The constituents present in the oils were quantified by peak-area normalization using a GC-FID instrument.

The essential oil of Pataqueira was first evaluated for its nature and stability in ethanol (PEO/E) at 1000 ppm, used as a control, and named S-1. Then, the oil was analyzed in the presence of hydrotalcite (PEO/E/LDH) at a weight ratio 1:1 and named S-2. The main constituents identified in these two oils were thymol (41.2% and 41.1%), thymol methyl ether (39.2% and 38.3%), α-phellandrene (11.6% and 12.1%), and *p*-cymene (1.6% and 1.6%), respectively, representing about 93% of the total composition (see Table 1). The composition of the oils (S-1 and S-2) was very similar to another Pataqueira oil (S-3) from fresh and dried plants, previously analyzed by Rebelo and colleagues (2009) [7] (see Table 1). Furthermore, preliminary tests found that the reaction catalyzed by the hydrotalcite was most favored in the presence of water (W). Based on these results, an oil stock solution was prepared in ethanol to evaluate the catalytic reactions and to compare them with the oil samples containing the synthesized hydrotalcite in the presence and absence of water (see item 2.3).

### 2.3. Catalytic Tests of the Oil with Water and Hydrotalcite

Six new oil samples were prepared based on the oil control sample (PEO/E, S1) solubilized in ethanol. These were three samples only with water, in the proportions (*v*/*v*) 5:1 (5PEO/E:1W, S-4), 2:1 (2PEO/E:1W, S-6), and 1:1 (1PEO/E:1W, S-8), and three other samples with water plus hydrotalcite, under the same proportions: (*v*/*v*) 5:1 (5PEO/E:1W + 1LDH, S-5), 2:1 (2PEO/E:1W + 1LDH, S-7), and 1:1 (1PEO/E:1W + 1LDH, S-9). Therefore, three samples of oil were mixed with varying proportions of water, and three samples of oil were mixed with different proportions of water plus hydrotalcite. As the percentages of thymol, thymol methyl ether, α-phellandrene, and *p*-cymene corresponded to more than 93% of the total composition of the oil, and these constituents are influenced by the same biogenetic pathways of the plant, the authors decided to evaluate only the variation in these oil main constituents, regarding the performance of catalytic tests with water and hydrotalcite. The results can be seen in Table 2 and Figure 3.

In the catalytic tests, with increasing amounts of water and hydrotalcite, the percentage of thymol methyl ether, α-phellandrene, and *p*-cymene decreased, while the percentage of thymol increased. The content of α-phellandrene was reduced from 20.4% to 1.9% and that of *p*-cymene from 3.1% to 0.6% (S-4 to S-9), respectively. Meanwhile, the thymol methyl ether content dropped from 56.9% (S-4) to 34.3% (S-9), and the thymol content increased from 13.1% (S-4) to 56.6% (S-9), respectively. That is, in the treatments (S4 to S9) with water dilution (activation) and catalytic action of hydrotalcite, the oil sample registered an increase of 43.5% in the thymol content, which may have been a consequence of the reduction in contents of thymol methyl ether (22.6%), α-phellandrene (18.5%), and *p*-cymene (2.5%), respectively. In addition to the hydrotalcite, the presence of water in the catalytic reaction observed was critical. Previously, in the study of Rebelo and co-workers (2009) [7] (see Table 1, S-3), the oils of Pataqueira obtained from the fresh (90% water) and naturally dried plants (10% water) showed that the thymol content was about 1.5 times lower in the oil of the dried plant (26.4%) than the oil of the fresh plant (40.0%), thus favoring the increase in the percentages of thymol methyl ether (47.7%), α-phellandrene (14.3%), and *p*-cymene (1.7%) in the dried plant. The plant drying process led to results that now can assist in interpreting the mechanism of action of water and hydrotalcite when reacting with the Pataqueira oil.

To ensure the best evidence for the action of water in the catalytic reaction with hydrotalcite for the conversion of the monoterpenes of Pataqueira, an oil sample was solubilized in deionized water at 1000 ppm concentration, which resulted in sample S-10 (PEO/W). Then, sample S-10 was divided into portions of 1:2 and 1:10 (oil/water–water, *v*/*v*), furnishing the samples S-12 (1PEO/W:2W) and S-14 (1PEO/W:10W). In turn, the samples S-10, S-12, and S-14 were added to the hydrotalcite in a ratio of 1:1, producing the samples S-11 (1PEO/W:1LDH), S-13 (1PEO/W:2W + 1LDH), and S-15 (1PEO/W:10W + 1LDH), respectively. So, the samples from S-10 to S-15 were analyzed by GC-FID and GC-MS, and the results can be seen in Table 3 and Figure 4.

Tests conducted with the samples S-10 to S-15 more clearly showed the conversion of the monoterpenes from Pataqueira oil, now accelerated by hydrotalcite in the presence of water. The percentage of thymol increased from 62.3% to 95.0%, with its highest value in the oil sample diluted twice with water (S-13) and catalyzed by the hydrotalcite. In the dilution of the oil sample by ten times (S-15), the percentage of thymol was only 87.2%, meaning there was a limit in the activation of the reaction with water. In general, the percentages of α-phellandrene and *p*-cymene were reduced from 20.4% and 3.1% (S-4) to zero (S-15), while the content of the thymol methyl ether decreased from 56.9% (S-4) to 0.3% (S-15). As previously mentioned, LDHs are promising materials for heterogeneous catalysis used effectively in various organic reactions [9,10,11,12]. It can be assumed that reactions of dehydrogenation and aromatization in the α-phellandrene led to the production of *p*-cymene that, in turn, suffered hydration and hydroxylation reactions to provide thymol. In addition, demethoxylation and hydration reactions may have occurred in the thymol methyl ether, which was also transformed in thymol (see Figure 5). The reaction with the hydrotalcite activated by water seems selective, producing hydroxyl derivatives such as thymol and *p*-cymen-8-ol. The latter is also present in the composition of the Pataqueira oil, whose percentage was increased in the catalytic reactions carried out with the samples S-10 to S-15.

In the biosynthetic pathway of monoterpenes, neryl and geranyl pyrophosphate are the natural substrates for the monoterpene synthases. All synthase enzymes can efficiently utilize them as precursors for the production of various metabolites, such as γ-terpinene, α-phellandrene, *p*-cymene, thymol, and thymol methyl ether, which are structurally related and occur in the Pataqueira oil. The co-occurrence of aromatic oxygenated monoterpenes, such as *p*-cymene, thymol, and derivatives, with cyclohexadiene-type monoterpene hydrocarbons which are structurally related, such as γ-terpinene and α-phellandrene, suggests the possibility that these metabolites must be biogenetically related within the same plant organism (see Figure 5). Incorporation studies of ^14^CO_2_ in volatile monoterpenes existing in cuttings of *Thymus vulgaris* showed strong evidence that the aromatization of α-phellandrene (or γ-terpinene) produces *p*-cymene, followed by a hydroxylation process leading to the biosynthesis of thymol [31,32,33,34].

Hydrotalcite has demonstrated its efficiency as a catalytic material and its high performance in directly producing thymol, which characterizes it as a highly selective catalyst. The results obtained with hydrotalcite agree with those published by Granger and colleagues (1964) [31]. These authors observed that the conversion of monoterpenes in plants could be achieved by non-enzymatic aromatization of γ-terpinene. Then, they proposed an autoxidative conversion of γ-terpinene to *p*-cymene, which acted as a critical intermediate for forming various aromatic monoterpenes.

## 3. Materials and Methods

### 3.1. Mineral Material

For the synthesis of the hydrotalcite, blast-furnace slag (BFS) from a steel mill located in Marabá, Pará state, Brazil, plus the following reagents of analytical grade: MgCl_2_·6H_2_O, NaOH and HCl, was used. The preparation method of the hydrotalcite was co-precipitation with increasing pH [35]. The experiment consisted of mixing an aqueous NaOH solution with an acid solution of BFS and MgCl_2_·6H_2_O at a theoretical molar ratio of Mg/Al 4:1. The BFS acid solution was obtained by digestion with HCl. The mixture was vigorously stirred at a temperature of 45 °C. The final pH was 11. The formed gel was subjected to hydrothermal treatment in a closed reactor at 100 °C for 19 h, followed by filtering, washing with deionized water, and drying at 100 °C for 12 h. The resulting solid was named BFS-LDH-4MgAl, or simply LDH.

### 3.2. Plant Material and Processing

Specimens of pataqueira were sampled in the experimental area of Empresa Brasileira de Pesquisa Agropecuária (Embrapa), Municipality of Belém, Pará State, Brazil. The plant was identified by comparison with an authentic voucher of *Conobea scoparioides* (MG174901) deposited in the herbarium of Emílio Goeldi Museum, Belém City, Pará State, Brazil. The fresh material (150 g) was submitted to hydrodistillation using a Clevenger-type apparatus (3 h). The oil was dried over anhydrous sodium sulfate, and its yield was calculated as a basis of the plant’s dry weight. The plant moisture content was calculated using an Infrared Moisture Balance for water loss measurement. The oil was codified as PEO.

### 3.3. Oil Composition Analysis

The oil was analyzed on a Thermo Electron DSQ II GC-MS instrument under the following conditions: DB-5ms (30 m × 0.25 mm; 0.25 μm film phase thickness) fused-silica capillary column; programmed temperature, 60–240 °C (3 °C/min); injector temperature, 250 °C; carrier gas, helium adjusted to a linear velocity of 32 cm/s (measured at 100 °C); injection type, split (1 μL of a 1:1000 hexane solution); split flow was adjusted to yield a 20:1 ratio; septum sweep was a regular 10 mL/min; EIMS electron energy, 70 eV; temperature of the ion source and connection parts, 200 °C. Regarding the volatile constituents, the quantitative data were obtained by peak area normalization using a FOCUS GC-FID operated under similar conditions to the GC-MS system, except for the carrier gas, which was nitrogen. GC run was replicated two times. The retention index was calculated for all the volatile constituents using an *n*-alkane (C_8_–C_32_, Sigma-Aldrich, St. Louis, MO, USA) homologous series [36].

### 3.4. Hydrotalcite Characterization 

Samples of BFS and LDH (BFS-LDH-4MgAl) were characterized by X-ray powder Diffraction (XRPD), Energy-Dispersive X-ray Spectroscopy (EDS), Scanning Electron Microscopy (SEM), and Infrared Spectroscopy (IRS). XRPD patterns of the samples for 2θ ranging from 5 to 75° were recorded on a Panalytical X’PERT Automatic Diffractometer with CuK_α_ radiation. The micrographs were generated in an SEM, LEO-1430 model (Carl Zeiss, Jena, Germany), with images obtained by secondary electrons, an electron beam current of 90 uA, constant acceleration voltage of 10 kV, and distance work of 12–15 mm. The samples were previously metalized with a thin layer of platinum. The ratio of Mg and Al was determined by EDS and held in conjunction with SEM analysis. IRS determined the carbonate anion intercalated in the LDH structure. The IR vibration spectra of samples were recorded on Thermo Electron Corporation equipment, IR 100 model [16].

### 3.5. Catalytic Reaction with the Oil and Hydrotalcite

The catalytic reactions were carried out in a batch reactor, where the oil (PEO), the solvent (EtOH and H_2_O, alone or mixed), and hydrotalcite (BFS-LDH-4MgAl) were loaded all at once at the temperature of 28 °C, for 24 h [17]. The variables were the solvents, their ratio (*v*/*v*), and the dilution factor (×). The weight ratio PEO:BFS-LDH-4MgAl was equal to 1. Then, the reaction mixture was centrifuged, and the supernatant was extracted with *n*-hexane (HPLC grade). After a further centrifugation step (3000 rpm, 20 min), the reaction product was analyzed by GC-FID and GC-MS.

## 4. Conclusions

The LDH synthesized from blast-furnace slag acted as a potent and selective catalyst in the reactions occurring with the monoterpenes of the Pataqueira oil. The XRPD analysis of LDH showed typical reflections of the hydrotalcite mineral, with a basal spacing of 7.87 Å, characteristic of the Mg-Al-Cl-CO_3_ system. The LDH was effective with its catalytic action in aqueous media, significantly increasing the thymol content and reducing the percentage of α-phellandrene, *p*-cymene, and thymol methyl ether in the Pataqueira oil, with a strong tendency towards the production of hydroxylated derivatives. The LDH behavior was similar to that of the hypothetical biosynthetic pathway, which could be expected for the output of the monoterpenes existing in the Pataqueira oil.

## Figures and Tables

**Figure 1 plants-13-01199-f001:**
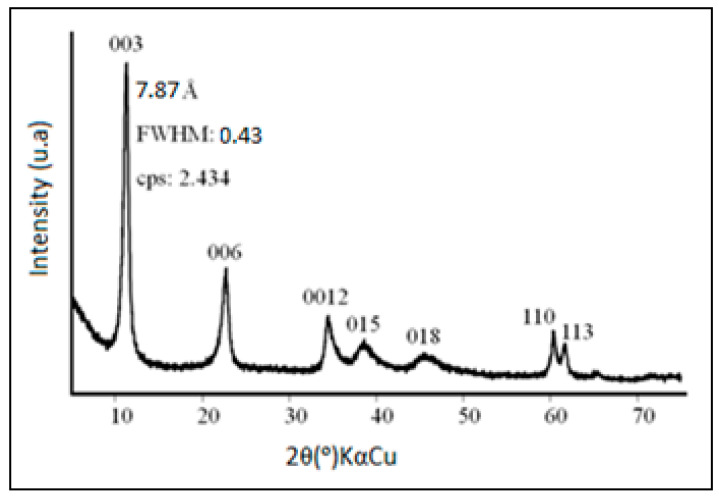
X-ray diffraction pattern of the hydrotalcite BFS-LDH-4MgAl, synthesized at 45 °C and pH 11.

**Figure 2 plants-13-01199-f002:**
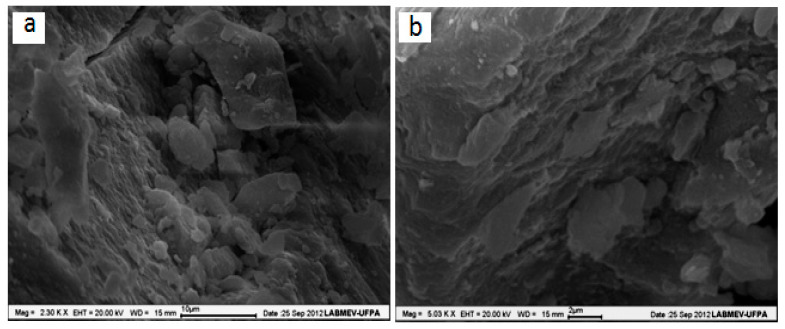
SEM micrographs of BFS-LDH-4MgAl with magnitudes 2.300× (**a**) and 5.030× (**b**) showing the morphology of typical lamellar porous material.

**Figure 3 plants-13-01199-f003:**
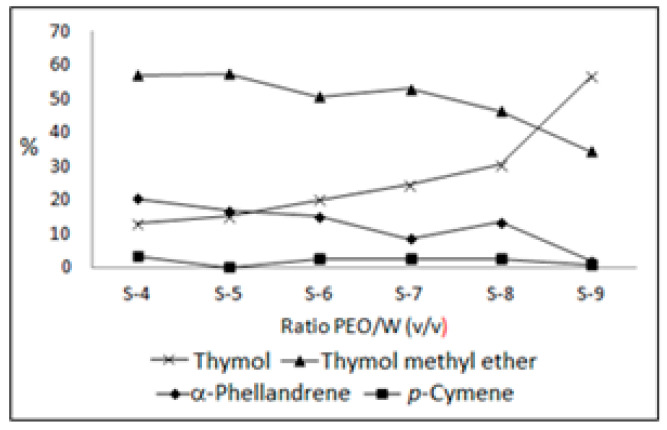
Percentage variation in the main constituents of the Pataqueira oil: the catalytic action of water and hydrotalcite in the oil–ethanol solution.

**Figure 4 plants-13-01199-f004:**
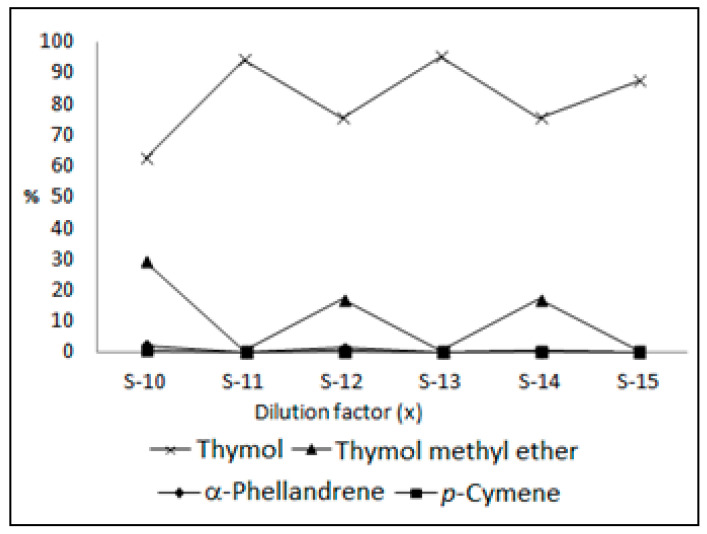
Percentage variation in the main constituents of the Pataqueira oil: the action of water and hydrotalcite in the oil.

**Figure 5 plants-13-01199-f005:**
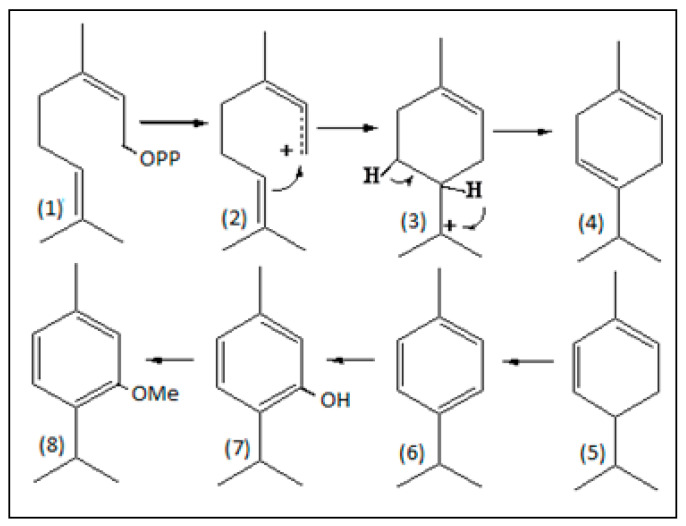
Principal constituents of the Pataqueira oil. Biosynthetic relationship originating from the precursor neryl pyrophosphate (1), intermediate carbocations (2,3), γ-terpinene (4), α-phellandrene (5), *p*-cymene (6), thymol (7), and thymol methyl ether (8).

**Table 1 plants-13-01199-t001:** Composition of volatile constituents identified in the Pataqueira oils.

Constituents	RI	S-1	S-2	S-3	
				Fresh Plant	Dried Plant
(*Z*)-3-Hexenol	858	0.1	0.2		
α-Thujene	930			0.2	0.3
α-Pinene	937	0.1	0.1		
Sabinene	975			0.2	
β-Pinene	979			0.3	0.2
3-Octanone	982	2.0	2.1		
α-Phellandrene	1002	11.6	12.1	12.1	14.3
*p*-Cymene	1020	1.6	1.6	1.5	1.7
Limonene	1028			0.1	0.1
β-Phellandrene	1030	0.5	0.5	0.2	0.2
(*E*)-β-Ocimene	1048		0.2	0.5	0.7
γ-Terpinene	1059	0.3	0.4	0.4	0.3
Linalool	1098	0.3	0.3	0.2	0.2
*cis*-*p*-Menth-2-en-1-ol	1120	0.1	0.1		
*allo*-Ocimene	1130	0.1	0.1		
Karahanaenone	1154	0.1	0.1		
*p*-Cymen-8-ol	1181	0.6	0.6	0.4	0.7
Thymol methyl ether	1232	39.2	38.3	39.6	47.7
Thymol	1290	41.2	41.1	40.0	26.4
Thymol acetate	1355	0.1	0.1		
Eugenol	1360	0.2	0.2	0.2	0.2
Viridiflorene	1496			0.5	1.2
α-Selinene	1498			0.2	0.9
(*E*,*E*)-α-Farnesene	1508	0.3	0.3	1.2	1.3
(*E*)-Nerolidol	1563	0.1	0.1		
Unidentified sesquiterpenes		0.2	0.3	0.4	1.4
Total (%)		98.7	98.8	98.2	97.8

RI = Retention Index (on DB-5ms column); S-1 = Pataqueira oil; S-2 = Pataqueira oil + hydrotalcite; S-3 = A previously analyzed Pataqueira oil [7].

**Table 2 plants-13-01199-t002:** Percentage variation in the main constituents of the Pataqueira oil: the catalytic action of water and hydrotalcite in the oil–ethanol solution.

Constituents	RI	S-4	S-5	S-6	S-7	S-8	S-9
*α*-Phellandrene	1002	20.4	16.8	15.2	8.5	13.5	1.9
*p*-Cymene	1020	3.1	2.8	2.5	2.3	2.3	0.6
*γ*-Terpinene	1059	0.8	0.7	0.6	0.7	0.5	0.1
*p*-Cymen-8-ol	1181	0.2	0.2	0.2	0.3	0.4	0.6
Thymol methyl ether	1232	56.9	57.2	50.5	52.9	46.3	34.3
Thymol	1290	13.1	15.0	20.0	24.4	30.3	56.6
Total (%)		94.5	92.7	89.0	89.1	93.3	94.1

RI = Retention Index (on DB-5ms column).

**Table 3 plants-13-01199-t003:** Percentage variation in the main constituents of the Pataqueira oil: the action of water and hydrotalcite in the oil.

Constituents	RI	S-10	S-11	S-12	S-13	S-14	S-15
*α*-Phellandrene	1002	2.0	0.1	1.3		0.3	
*p*-Cymene	1020	0.5	0.1	0.4		0.2	
*γ*-Terpinene	1059	0.1		0.1			
*p*-Cymen-8-ol	1181	0.8	1.4	0.9	1.3	0.9	1.1
Thymol methyl ether	1232	29.2	0.5	16.9	0.4	16.9	0.3
Thymol	1290	62.3	94.2	75.4	95.0	75.5	87.2
Total (%)		94.9	96.3	95.0	96.7	93.8	88.6

RI = Retention Index (on DB-5ms column).

## Data Availability

Data are contained within the article.

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
