# Peer review of "Phenolic Monoterpenes Conversion of Conobea scoparioides Essential Oil by Hydrotalcite Synthesized from Blast-Furnace Slag"

_plants, 2024, doi:10.3390/plants13091199_

Round 1

Reviewer 1 Report

Comments and Suggestions for Authors

This manuscript mainly reported the conversions of monoterpenes from the plant essential oil by using hydrotalcite as the catalyst. The monoterpenes were characterized by GC-MS, while the catalyst hydrotalcite was synthesized and further characterized by SEM and X-ray analysis. Further, the reaction was performed under different conditions, and concluded that hydrotalcite successfully catalyzed the conversion and accumulation of thymol from its precursors. As far as I am concerned, this manuscript can be assigned as minor revision. My only question is regarding to the reaction conditions. The variables are the solvents, their ratio (v/v), and the dilution factor. If temperature and amount of the catalyst are also set as the variables, how the results would be? The thymol methyl ester will be the major product, instead of thymol?

Reviewer 2 Report

Comments and Suggestions for Authors

The text sent for review concerns the issue of obtaining thymol, which is important in the light of broadly understood "green chemistry". The Authors showed that the species Conobea scoparioide produces an oil that can be easily modified to be an efficient source of thymol.

I have no reservations about the work carried out and the results obtained, but I do have strong reservations about the prepared text.

The Authors use a large number of abbreviations, which they freely mix with the full names, which often makes the text incomprehensible.

The rule of thumb is to first at beginning clearly define all the abbreviations you and then use them consistently.

Descriptions of the preparation of S1-S15 samples should be clearly described in one place, I suggest that it should be a table, then Figures 3 and 4 would also become clear.

Tables 1, 2 and 3 use the abbreviation IR, which usually stands for InfraRed spectrometry data, but should be RI  as retention index.

Line 207 mentions studies using 14CO2, when it is likely that 13CO2 was not radioactive.

In the list of literature, item 18, the name of the first author is Rachwalik, not "Rachwalick".

While the main part of the text is terribly written and needs to be rewritten, Materials and Methods is acceptable.

To sum up, the text needs to be significantly improved and only then can it be accepted for printing.

Round 2

Reviewer 2 Report

Comments and Suggestions for Authors

The revised version of the article is more readable than the previous one. The Authors didn't follow my suggestion to tabulate the changes in the preparation of subsequent samples, which would make reading and understanding much easier, but it's still good. In my opinion, the version presented today can be published.

Author Response

Thank you for your review.